# PARTIALLY MUTUAL EXCLUSIVE SOFTMAX FOR POSITIVE AND UNLABELED DATA

## ABSTRACT

In recent years softmax, together with its fast approximations, has become the de-facto loss function for deep neural networks with multiclass predictions. However, softmax is used in many problems that do not fully fit the multiclass framework and where the softmax assumption of mutually exclusive outcomes can lead to biased results. This is often the case for applications such as language modeling, next-event prediction, and matrix factorization, where many of the potential outcomes are not mutually exclusive, but are more likely to be conditionally independent given the state. To this end, for the set of problems with positive and unlabeled data, we propose *Partially Mutual Exclusive Softmax (PMES)*, a relaxation of the original softmax formulation, where, given the observed state, each of the outcomes are conditionally independent but share a common set of negatives. Since we operate in a regime where explicit negatives are missing, we create a cooperatively-trained model of negatives, and derive a new negative sampling and weighting scheme which we call *Cooperative Importance Sampling (CIS)*. We show empirically the advantages of our newly introduced negative sampling scheme by plugging it into the Word2Vec algorithm and evaluating it extensively against other negative sampling schemes on both language modeling and matrix factorization tasks, where we show large lifts in performance.

## 1 INTRODUCTION

Learning from positive and unlabeled data is a well-defined task in machine learning, known as positive-unlabeled (PU) learning (Elkan & Noto (2008); Li & Liu (2003); Lee & Liu (2003); Ren et al. (2014); Paquet & Koenigstein (2013)). The applications of PU learning are numerous, as in many fields negative data is either too expensive to obtain or too hard to define. This is the case in language modeling, where one only observes examples of valid sentences and documents, and tries to learn a generative process. Similarly, in computer vision, with the recent work on Generative Adversarial Networks (GANs), one tries to learn the underlying generative process that produces meaningful images. In both cases, the space of negatives (e.g., non-sentences or non-images) is not clear. Similarly, in matrix factorization, most of the matrices contain pairs of observed interactions, and there are no available explicit negatives.

In all of the applications enumerated above, it is quite common to encounter solutions that are based on a softmax loss function. In language modeling, and more recently in matrix factorization, the *Word2Vec* algorithm (see Mikolov et al. (2013a)) models the conditional probability of observing all possible items in the vicinity of a context item as a categorical distribution using the softmax loss.

However, modeling the probability of co-occurrence as a categorical distribution is a very biased assumption that is clearly refuted by the data, since for all context words there are multiple words that co-occur at the same time.

In our paper we propose *Partially Mutual Exclusive Softmax (PMES)*, a new model that relaxes the mutual exclusivity constraint over the outcomes. PMES relaxes this constraint by splitting the set of all outcomes into a set of possible outcomes that are conditionally independent given the context, and a set of impossible outcomes which become negative examples.

The context-dependent negative set is hypothesized but not known, so in our method we introduce a model for negatives that is used to weight the sampled candidate negatives. The training algorithm is based on the simultaneous training of two neural networks. The first network is a generator that fits the positives and the sampled negatives. The second network is the discriminator, which is trained to separate the true positive pairs from generated pairs, and is used as the model of the probability that an example would receive a negative label.

The resulting solution has many similarities with other recent negative sampling methods for approximating full softmax. However, unlike most of the previous methods, our method is not trying to faithfully approximate the full softmax formulation, but to fix some of its over-simplifying assumptions. Furthermore, we believe that the observed lift in performance of some of the negative sampling work over the full softmax can be explained through the prism of *Partially Mutual Exclusive Softmax*.

Our hypothesis is further confirmed by experiments on language modeling and matrix factorization, where we show a big lift in performance over previous work on negative sampling and full softmax. We validate some of our intuitions on the advantages of our sampling procedure on an artificial dataset where the support sets are known, and show that our sampling scheme correctly approximates the support, which is not the case for other softmax variants.

Overall, the main contributions of this paper are the following:

- We propose *Partially Mutual Exclusive (PME) Softmax*, a modified version of the softmax loss that is a better fit for problems with positive and unlabeled data.

- We derive a new negative sampling scheme based on an cooperatively-trained models of negatives which we denote as *Cooperative Importance Sampling (CIS)*

- We show empirically the validity of our proposed approach by plugging our new loss into the Word2Vec model, and evaluating this enhanced Word2Vec model against classical sampling schemes for Word2Vec on language modeling and matrix factorization tasks across six real-world datasets.

We discuss related work on Word2Vec, negative sampling schemes for softmax and GANs in Section 2 of this paper. In Section 3 we formally introduce our *PME-Softmax* loss and the associated CIS negative sampling scheme, and describe the training algorithm. We highlight the performance of our method in Section 4, and conclude with ideas and directions for future work in Section 5.

## 2 RELATED WORK

### 2.1 WORD2VEC, SOFTMAX AND NEGATIVE SAMPLING SCHEMES

With the introduction of Word2Vec, Mikolov et al. (2013b;a) achieved state-of-the-art results in terms of the quality of the words embeddings for various language modeling tasks. In the last couple of years, Word2Vec became a widely embedding method, that is used in various setups from language modeling to matrix factorization. Since then, it is important to note the progress made in improving words embeddings (Bojanowski et al. (2016); Pennington et al. (2014)), especially more recently with the FastText model (Joulin et al. (2016)) that also leverages n-gram features.

When dealing with the training of multi-class models with thousands or millions of output classes, one can replace softmax by candidate sampling algorithms. Different methods can speed up the training by considering a small randomly-chosen subset of candidates for each batch of training examples. Gutmann & Hyvärinen (2010) introduced Noise Contrastive Estimation (NCE) as an unbiased estimator of the softmax loss, and it has been shown to be efficient for learning word embeddings Mnih & Teh (2012). In Mikolov et al. (2013b), the authors propose a *negative sampling* loss not to approximate softmax and but to learn high-quality vectors. Defined in Bengio et al. (2003), *sampled softmax* can also be used to train these language models. Benchmarked in ((Bengio & Senécal, 2008; Jean et al., 2014)), the authors have have shown state-of-the-art results in terms of performance and time complexity. Indeed, sampled softmax avoids computing scores for every possible continuation. Here, one chooses a proposal distribution from which it is cheap to sample, and performs a biased importance sampling scheme for approximating the softmax.

Most previous work has focused on generic sampling methods such as uniform sampling and unigram sampling. More recently, in Chen et al. (2018), the authors provide an insightful analysis of negative sampling. They show that negative samples that have high inner product scores with the context word are more informative in terms of gradients on the loss function. They show analytically and empirically the benefits of sampling from the unigram distribution. Leveraging this analysis, the authors propose a dynamic sampling method based on inner-product rankings.

## 2.2 GANs

First proposed by in 2014 in Goodfellow et al. (2014), GANs have been quite successful at generating realistic images (Ledig et al. (2016); Radford et al. (2015); Miyato et al. (2018)). GANs can be viewed as a framework for training generative models by posing the training procedure as a minimax game between the generative and discriminative models.

Our approach leverages recent work on discrete GANs. Bose et al. (2018) describes an approach for adversarial contrastive estimation, where discrete GANs are leveraged to implement an adversarial negative sampler. However, our approach differs as in our method, called *cooperative importance sampling*, both the generator and the discriminator are trained to work cooperatively.

## 3 OUR APPROACH

### 3.1 CONCEPTS AND NOTATION

We begin this section by formally defining the task. We denote by $I$ and $J$ two sets of objects. These objects can potentially represent sets of users, items or words. Given an $i \in I$ sampled uniformly from $I$, we denote by $P(.|i)$ the unknown conditional generative process that generates the set of $j \in J$ and by $X$ the resulting joint distribution over $I \times J$. In our data, we assume that we only observe a finite sample of pairs $(i, j) \sim X$.

We denote $\mathcal{G} = \{G_\theta\}_{\theta \in \Theta}$ as the family of functions, with parameters $\Theta \subset \mathbb{R}^p$. The model $G_\theta$ takes an input in $I \times J$ and outputs a score $G_\theta(i, j) \in \mathbb{R}$. The objective is to learn a generative model $G_\theta$ that fits the conditional process observed in the data.

### 3.2 SOFTMAX ASSUMPTION : MODELING THE DATA AS A 1D RANDOM VECTOR

Usually, one tries to model the process $X$ with a softmax formulation (see eq 1), defining a multinomial distribution $P(.|i) \in P(J)$. Therefore, for a given context $i$ and any two targets $(j_1, j_2)$, $P(j_1|i)$ and $P(j_2|i)$ are not independent since the sum of all conditional probabilities over the target set must sum to 1. That means, that under this formulation, the probability of observing $j_2$ given we already observed $j_1$ in the context of $i$ goes to zero. We denote this property *mutual exclusivity*.

$$\forall \mathbf{i} \in I, \ g_\theta(j|\mathbf{i}) = \frac{\exp(G_\theta(i,j))}{\exp(G_\theta(i,j)) + \sum_{j' \in J \setminus j} \exp(G_\theta(i,j'))} \tag{1}$$

In order to fit this model, for a given $i \in I$, one tries to perform Kullback-Leibler divergence minimization between the true distribution $P(.|i)$ and the model of the distribution $g_\theta$:

$$\underset{\theta \in \Theta}{\mathrm{argmin}} \ KL(P_i||g_\theta(.|i)) = \underset{\theta \in \Theta}{\mathrm{argmax}} \ \mathbb{E}_{j \sim P_i} \ [\ln(g_\theta(j|i))] \tag{2}$$

$$\underset{\theta \in \Theta}{\mathrm{argmax}} \ \mathbb{E}_{j \sim P_i} \ G_\theta(i,j) - \ln \sum_{j' \in J} \exp(G_\theta(i,j')) \tag{3}$$

In the case of multi-class and single-label tasks it is natural to use the softmax formulation. However, when it comes to language modeling and sequence prediction, most of the tasks fall in the multi-labeled settings. For a given context $i$, one does not observe one target item $j$ but rather a subset of target items $\{j_1, ..., j_k\} \in J^k$. For example, in a *word2vec* setting, the subset of targets is defined by the *sliding window* parameter. We generalize the current formulation to the multi-labeled settings as

follows:

$$P(\{j_1..j_k\}|i) = \prod_{x=1}^{k} g_\theta(j_x|i), \text{ with} \sum_{j \in J} g_\theta(j|i) = 1 \tag{4}$$

Inspired from textual analysis, Ueda & Saito (2003); Blei et al. (2003) suggested that words in sequences can be regarded as a mixture of distributions related to each of the different categories of the vocabulary, such as "sports" and "music". Building upon this example, we effectively search over an enlarged class of models to better represent the *multiplicity* of the data. As detailed in the next subsection, we now train a product of independent distributions to learn this set generative process.

### 3.3 PME SOFTMAX : MODELING THE DATA AS A N-DIMENSIONAL RANDOM VARIABLE

We propose to directly model the joint conditional distribution over a set of target items. More formally, we now model the output $P(.|i)$ as a $n$-dimensional ($n = \mathbf{card}(J)$) random vector of Bernoulli random variables, each with a parameter $g_\theta(j|i)$. Consequently, for a given set $\{j_1, ..., j_k\} \in J^k, k \leqslant n$, we have:

$$P(\{j_1..j_k\}|i) = \prod_{x=1}^{k} g_\theta(j_x|i) \text{ with no constraint on} \sum_{j \in J} g_\theta(j|i) \tag{5}$$

The problem now is to find the right parameter $g_\theta(j|i)$ of each Bernoulli variable. We assume that :

$$\forall (i,j) \in I \times J, \ g_\theta(j|i) = \frac{\exp(G_\theta(i,j))}{\exp(G_\theta(i,j)) + \sum_{j' \in \neg j} \exp(G_\theta(i,j'))} \tag{6}$$

where $\neg j$ is the set of items mutually exclusive with $j$ given $i$.

It is not obvious how to define the set $\neg j$. Softmax assumes that all other events are mutually exclusive and we would like to relax this assumption.

**Partially Mutual Exclusive Softmax** For a pair $(i,j)$, only a subset of words are very unlikely to co-occur with $i$ given that the target $j$ already co-occurs with $i$. For example, we assume that words that share the same *grammatical category* or *semantic context* with $j$ might not co-occur with $i$. As a first version, we simplify this model by assuming that for every context $i \in I$, there is a set of negatives that will not co-occur with $i$ independently of the observed target word $j$. This means that we model $g_\theta(j|i)$ with the following *PME-Softmax* formulation:

$$\forall (i,j) \in I \times J, \ g_\theta(j|i) = \frac{\exp(G_\theta(i,j))}{\exp(G_\theta(i,j)) + \sum_{j' \in \neg S_i} \exp(G_\theta(i,j'))} \tag{7}$$

where $S_i$ is defined as the true but unknown support set of items $j$ possibly co-occuring with $i$. Contrary to the softmax formulation, for all items $j$ in $S_i$, each Bernoullis with parameter $g_\theta(j|i)$ are independent.

In this formulation, the normalization factor is computed over the positive pair $(i,j)$ and the sum of the probabilities over the negative pairs. However, since in reality we do not have access to the support $S_i$, we replace it with a probabilistic model $D_i$. In this way we replace the exact formulation of the *PME-Softmax* with a probabilistic one, where all of the $j'$ items sampled are weighted by the probability $D_i(j)$ of being in the negative set:

**Probabillistic Partially Mutual Exclusive Softmax**

$$\forall (i,j) \in I \times J, \ g_\theta(j|i) = \frac{\exp(G_\theta(i,j))}{\exp(G_\theta(i,j)) + \sum_{j' \in V_i} \exp(G_\theta(i,j')) \times D_i(j')} \tag{8}$$

where $D_i$ is the probabilistic model of $\neg S_i$ and $V_i$ is a given subset of targets. This loss is very close to the Sampled Softmax loss defined in Bengio et al. (2003) but is not an approximation of softmax.

### 3.4 THE COOPERATIVE GAME

In order to have a model of these negatives, we leverage recent work on GANs by training simultaneously a generator and a discriminator.

We denote $\mathscr{D} = \{D_\alpha\}_{\alpha \in A}$ as the family of functions, with parameters $A \subset \mathbb{R}^p$. The model $D_\alpha$ takes an input $(i, j) \in I \times J$ and outputs a score $D_\alpha(i, j) \in \mathbb{R}$. $D_\alpha$ is be trained with a binary cross-entropy loss function (BCE), to discriminate between positive data sampled from $X$ and negatives sampled by our generator.

$$\underset{\alpha \in A}{\text{argmax}} \ \mathbb{E}_{(i,j) \sim X} \ \ln \ \sigma(D_\alpha(i, j)) + \sum_{j=1}^{k} \mathbb{E}_{j \sim g_\theta(.|i)} \ [\ln \ \sigma(-D_\alpha(i, j))] \tag{9}$$

where $\sigma$ is the sigmoid function.

In our sampling model, for a given $i \in I$, we use the modeled softmax conditional distribution $g_\theta(.|i) \in P(J)$ to sample negatives. It is true that sampling with $g_\theta$ requires computing the score for every possible continuation, but it is an effective way to sample negatives that are close to the decision boundary. However, as the model improves during training, the likelihood of sampling positives gets higher. To fix this, $D_\alpha$ is used in a second step, to oversample the samples that are true negatives. This model called *cooperative importance sampling*, has the following training loss:

$$\underset{\theta \in \Theta}{\text{argmax}} \ \mathbb{E}_{(i,j) \sim X} \ G_\theta(i, j) - \ln \left( \sum_{j \in V} \exp(G_\theta(i, j)) \times \sigma(-D(i, j_k)) \right) \tag{10}$$

where $V := \{j\} \bigcup \{j_1, ..., j_n\}$, $j_1, ..., j_n \sim g_\theta(.|i)$.

**Improved gradient via $g$ and $D$** Intuitively, we want to find true hard negatives that will be informative in terms of gradient. Ideally, we would only sample negatives that are mutually exclusive with $j$ given the context $i$. When deriving the sampled softmax loss, we have the following gradient:

$$\nabla G_\theta(i, j) - \sum_{(i,j) \in V} p_{\theta,\alpha}\big|_V(j|i) \ \nabla G_\theta(i, j) \tag{11}$$

where $p_{\theta,\alpha}\big|_V(j|i) = \frac{\exp(G_\theta(i,j)) \times D_\alpha(i,j)}{\sum_{j \in V} \exp(G_\theta(i,j)) \times D_\alpha(i,j)}$, a distribution whose support lies in $V$.

We see that if the negatives sampled are too easily distinguishable from real data, then they will be far from the decision boundary, have low gradients and therefore will not enable $G_\theta$ to improve itself. Consequently, our sampling method guarantees to have meaningful samples, and $D$ ensures us to oversample the true negatives in $V$.

### 3.5 ALGORITHM

A description of our algorithm can be found in Algorithm 1. We empirically show improvements with this procedure in the following section.

## 4 EXPERIMENTS

We compare our *CIS* negative sampling scheme against full softmax and the negative schemes listed below on three types of datasets, namely a 2D artifical dataset where we know the true support, a text datasets on which we evaluate word ranking, word similarity and word analogy and five matrix factorization datasets.

The different baselines are the following: Full Softmax where one uses for every context all items as negatives (FS), Uniform sampling where the negatives are sampled uniformly (UniS), Popularity sampling where the sampling distribution is a log-uniform distribution based on popularity ranking (PopS). The Selfplay method (SP) is defined by sampling the negative samples in the softmax distribution $g_\theta$ of our model like in our *cooperative importance sampling scheme* except that we do not correct the weights with the help of the discriminator $D_\alpha$. Sampling within its own distribution

---

**Algorithm 1** AIS

---

**Require:** generator $G_\theta$; discriminator $D_\alpha$; a generative process $X$.
  Initialize $G_\theta$, $D_\alpha$ with random weights $\theta, \alpha$.

  **repeat**
    **for** g-steps **do**
      Sample pairs $(i, j) \sim X$
      Generate samples from the softmax distribution $g_\theta(.|i)$
      Train the generator $G_\theta$ with Eq. (10)
    **end for**
    **for** d-steps **do**
      Sample pairs $(i, j) \sim X$
      Generate samples $(j_1, ..., j_n)$ from $g_\theta(.|i)$
      Train the discriminator $D_\alpha$ with Eq. (9)
    **end for**
  **until** AIS converges

---

| Method | Blobs ($r_1 = 0.25$) | Blobs ($r_2 = 0.60$) | Blobs ($r_3 = 0.75$) | Swiss Roll |
|--------|--------|--------|--------|--------|
| FS | $69.0 \pm 0.2$ | $77.0 \pm 0.2$ | $84.5 \pm 0.3$ | $89.1 \pm 0.1$ |
| UniS | $72.3 \pm 0.1$ | $70.1 \pm 0.1$ | $75.1 \pm 0.2$ | $87 \pm 0.2$ |
| PopS | $75.2 \pm 0.2$ | $65.0 \pm 0.1$ | $77.2 \pm 0.1$ | $88.2 \pm 0.1$ |
| Selfplay | $75.7 \pm 0.3$ | $76.5 \pm 0.1$ | $84.5 \pm 0.1$ | $88.5 \pm 0.1$ |
| CIS | $\mathbf{78.5 \pm 0.1}$ | $\mathbf{86.1 \pm 0.2}$ | $\mathbf{89.5 \pm 0.1}$ | $\mathbf{90.2 \pm 0.1}$ |

Table 1: Table showing the Accuracy results on the synthetic datasets with different ratios $(r_1, r_2, r_3)$ of positive data. CIS outperforms other models on every dataset but by a smaller margin when the underlying manifold is easier to learn like in the Swiss-Roll dataset.

allows us to sample negatives close to the decision boundary. However, without $D$'s reweighting, we expect the Selfplay method to perform worse than the CIS model.

Across all experiments, embedding size is kept fixed at 150. Also, we always hold out 20% of the dataset for the test. The MPR refers to the standard Mean Percentile Rank as defined and, Prec@k refers to the fraction of instances in the test set for which the target falls within the top-k predictions.

### 4.1 LEARNING 2D DISCRETE DISTRIBUTION: SYNTHETIC DATASETS

To verify our intuition around true support modeling, we defined a set of 2D artificial datasets where all positive pairs live in 2D shapes and the task of the Word2Vec is to learn these shapes from samples of positives and unlabeled data. The 2D shapes we experimented with are the *Swiss Roll*, the *S Curve* and overlapping blobs of various size. An advantage of the blobs-based simulations is that they allow us to better control the ratio $r$ of positive data. One can look in the Appendix, where we present different figures showing how well the different methods learn the underlying distribution.

We report the accuracy of our predictions in the Table 1 below. We observe that our cooperative importance sampling is systematically better than other sampling methods and on par with full softmax. Furthermore, it is interesting to note that when the underlying distribution is quite easy to learn the impact of CIS is smaller, as $D$ cannot fully correct the sampling of positive data.

### 4.2 LANGUAGE MODELING TASKS

For the language modeling tasks, we ran experiments on the *text8* dataset, a textual data excerpt from Wikipedia taken from *Matt Mahoney's* page. We ran two experiments where we only kept respectively the 12,000 and 30,000 most-occuring words in the data and ran Skip-Gram Word2Vec with a window size 1 (after removing the rare words) and the versions of negative sampling listed above. We ran experiments on threee different tasks: next word prediction, similarity task and analogy task. We also compared to the FastText (FT) model implemented by Joulin et al. (2016), which is considered to provide state-of-the-art results. We trained it on the same data and benched on different parameters.

**Next word prediction** In the next word prediction task, we use the model to predict the target word for a given context. We report both the Mean Percentile Rank (MPR) and Precision@1 metrics. We see that on these two metrics our model beats the different sampling methods, including full softmax.

| Method | Text8 (12000) | | Text8 (30000) | |
|---|---|---|---|---|
| FS | $85.1 \pm 0.08$ | $7.2 \pm 0.15$ | $89.4 \pm 0.08$ | $7.5 \pm 0.15$ |
| UniS | $87.4 \pm 0.07$ | $6.0 \pm 0.12$ | $90.6 \pm 0.07$ | $5.5 \pm 0.17$ |
| PopS | $87.6 \pm 0.06$ | $7.2 \pm 0.13$ | $90.7 \pm 0.06$ | $7.1 \pm 0.18$ |
| Selfplay | $88.2 \pm 0.04$ | $8.9 \pm 0.10$ | $88.7 \pm 0.05$ | $8.6 \pm 0.16$ |
| CIS | $\mathbf{88.7 \pm 0.08}$ | $\mathbf{8.9 \pm 0.10}$ | $\mathbf{91.6 \pm 0.08}$ | $\mathbf{9.3 \pm 0.09}$ |

Table 2: Table showing the MPR and P@1 for the different models on the two text datasets. We see that even using the selfplay method helps us sample hard negatives.

**The words similarity task** In terms of qualitative results, we show in Table 3 a set of 6 query words and their 6-nearest neighbours in terms of the cosine distance in the embedding space in the 30k different words dataset. We can see that the full softmax and our cooperative sampling method have some similarities in terms of results, except that softmax sometimes lacks coherence, especially as we increase this number of neighbors. On the other hand, FastText and our importance cooperative sampling tend to have very different results. More similarities between words are shown in Appendix.

| Method | CIS | FastText | Full Softmax |
|---|---|---|---|
| assembly | senate, legislature
parliament, seats
cabinet, democracy | assembl, assemblies
assemble, assembling
assembler, assembled | government, system
union, party
council, parliament |
| author | poet, writer
composer, historian
philosopher, novelist | authored, writer
authors, novelist
authorship, poet | writer, actor
scholar, poet
artist, actress |
| australia | canada, scotia
usa, indonesia
bulgaria, argentina | australis, australiasia
australians, australasian
zealand, queensland | island, canada
day, georgia
kingdom, party |
| true | meaning, essence
false, truth
sense, existence | untrue, truth
false, truths
theistic, falsehood | meaning, case
possible, definition
or, understanding |
| essay | bibliography, thinking
essays, prophecy
story, poetry | essays, essai
essayist essayists
critique, treatises | treatise, downloads
interviews, mentor
essays, pupil |
| final | previous, latest
next, first
last, second | finale, finals
finalized, penultimate
last, eventual | previous, last
first, another
initial, second |

Table 3: In this table, we show qualitative results for the Similarity task results. We can see that FastText focuses more on N-grams similarities and shows less diversity.

**The word analogy task** The goal of the word analogy task is to compare the different embedding structures. With the pair based method, given word $i$ and its analogy $j$, one has to find, given a third word $i'$, its analogy $j'$. Usually, we consider two types of analogy tasks: *semantic* and *syntactic* analogies. To test analogies, we used the Google analogy test set developed by Mikolov et al. (2013b). Semantic analogies test the robustess of our embeddings to transformations like "man" to "woman" while syntactic analogies focus on transformations like "run" to "runs". In terms of quantitative results we show the Prec@1, Prec@5 and Prec@15 in the Table 6 of *CIS* against the baseline models and see a clear uplift in terms of quality. As FastText relies on the n-gram information, they manage to beat CIS on the syntactic analogy task, especially on the 30k different words dataset.

### 4.3 Matrix Factorization Experiments : Shopping Baskets and Movies datasets

For the matrix factorization application we performed our experiments on four different public datasets: two shopping datasets listing products purchased together (the *Belgian Retail* dataset and the

12,000 most-occuring words

| Method | CIS | SP | FT | FS |
|---|---|---|---|---|
| Semantic | | | | |
| P@1 | **14.8** | 9.3 | 0.0 | 1.1 |
| P@5 | **19.7** | 14.3 | 4.4 | 7.1 |
| P@15 | **21.4** | 18.7 | 17.0 | 11.0 |
| Syntactic | | | | |
| P@1 | 1.1 | **1.7** | 0.6 | 0.1 |
| P@5 | 4.2 | **6.8** | 1.7 | 1.7 |
| P@15 | 6.8 | **10.2** | 9.3 | 3.8 |

30,000 most-occuring words

| Method | CIS | SP | FT | FS |
|---|---|---|---|---|
| Semantic | | | | |
| P@1 | **14.75** | 10.8 | 1.9 | 10.7 |
| P@5 | 20.6 | **22.2** | 17.0 | 16.0 |
| P@15 | 35.8 | 35.3 | **41.17** | 21.9 |
| Syntactic | | | | |
| P@1 | 1.9 | 2.7 | **8.9** | 1.2 |
| P@5 | 7.15 | 8.9 | **26.7** | 5.1 |
| P@15 | 11.80 | 16.1 | **46.0** | 8.7 |

Table 6: On this table we present the results for the Analogy task on respectively the 12k and 30k different words datasets. On both datasets, CIS/SP are ahead in the Semantic tasks on the Prec@1/5 metrics. However, when increasing the number of Nearest Neighbors, FastText catches up which might indicate a better structure in the tail.

*UK retail* dataset with respectively 16,470 and 4,070 different items) and two movie recommandation datasets (the *Movielens* and the *Netflix* datasets) listing movies seen by the same user. For the movie datasets, we only kept the movies ranked over respectively 4 and 4.5 stars keeping therefore 17,128 and 17770 movies. From these, we create positive only co-occurence datasets from all the possible pair combinations of items. Therefore, we transform this into an implicit task where we only have access to positive data.

On these datasets, we report the Prec@1 on the test data as shown in Table 7. CIS outperforms all baselines and gets relatively better when the vocabulary size of the dataset increases.

| Method | Belgian | UK | Movielens | Netflix |
|---|---|---|---|---|
| FS | $10.8 \pm 0.1$ | $2.7 \pm 0.1$ | $2.2 \pm 0.1$ | $2.2 \pm 0.1$ |
| UniS | $10.2 \pm 0.2$ | $2.4 \pm 0.1$ | $1.4 \pm 0.3$ | $1.4 \pm 0.2$ |
| PopS | $10.6 \pm 0.1$ | $2.4 \pm 0.2$ | $2.5 \pm 0.1$ | $1.4 \pm 0.2$ |
| Selfplay | $10.8 \pm 0.1$ | $2.5 \pm 0.1$ | $2.5 \pm 0.1$ | $2.2 \pm 0.1$ |
| CIS | $\mathbf{11.2 \pm 0.2}$ | $\mathbf{3.1 \pm 0.1}$ | $\mathbf{3.7 \pm 0.1}$ | $\mathbf{2.6 \pm 0.1}$ |

Table 7: This table shows the P@1 for the 4 different models (Full Softmax, Uniform Sampling, Popularity Sampling, Selfplay and CIS) on the real datasets on the Item-Items task. On the task of Prec@1, CIS consistently outperfoms the full softmax.

## 5 CONCLUSIONS

In this paper, we have proposed *Partially Mutual Exclusive Softmax*, a relaxed version of the full softmax that is more suited in cases with no explicit negatives, e.g., in cases with positive and unlabeled data. In order to model the new softmax we proposed a cooperative negative sampling algorithm. Based on recent progress made on GANs in discrete data settings, our cooperative training approach can be easily applied to models that use standard sampled softmax training, where the generator and discriminator can be of the same family of models. In future work we will investigate the effectiveness of this training procedure on more complex models, and also try to make our mutually exclusive set model more contextual and dependent on both objects $i$ and $j$ within a pair. For example, for a given pair context/target, one might want to use the closest neighbors of the target in the embedding space as negatives. This could enable us to obtain a negative distribution that fits both the context and the target.

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

## A EXPERIMENTS

### A.1 LEARNING 2D DISCRETE DISTRIBUTION

Here we show the different heatmaps regarding the different conditional models learned by the model on 4 different methods: Popularity sampling, Selfplay sampling, Full Softmax and our Cooperative Importance Sampling. See Image 1 and Image 2.

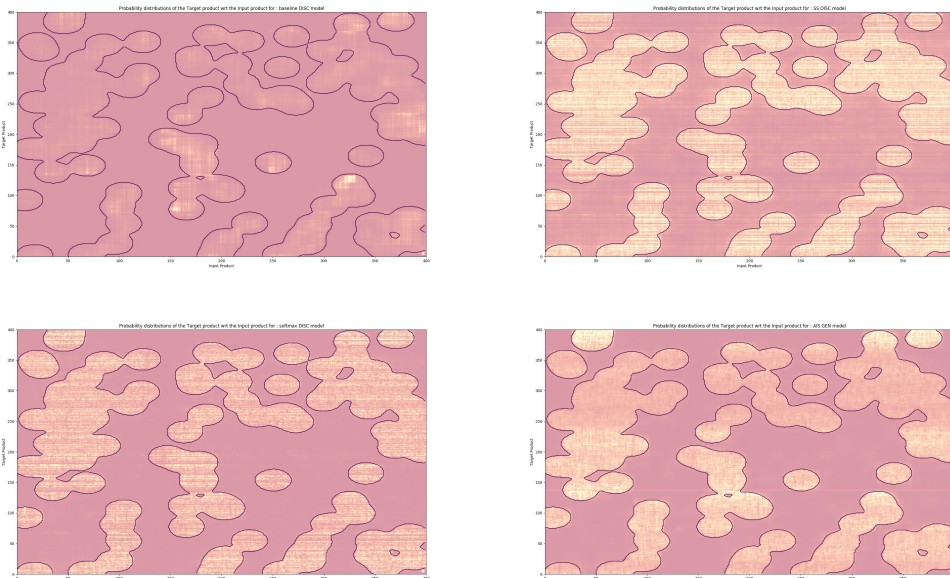

Figure 1: Learning the conditional distribution for one of the blobs dataset $r_2 = 0.6$. From the top left to the bottom right: the Popularity based sampling, the Selfplay model (top left) the Softmax and the CIS model. We see the impact of the CIS model on the learning of the right conditional distribution as the distinction between true data (inside the blobs) and negative data is clearer.

### A.2 THE WORDS SIMILARITY TASK

Here we present some other similarities between words generated from three different methods: Full Softmax, FastText and our CIS model. See Figure 8.

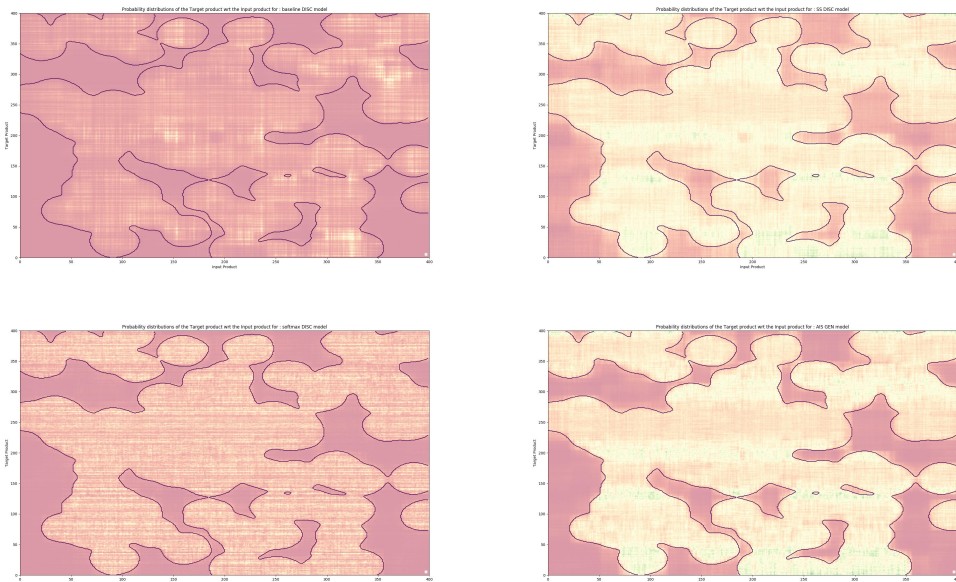

Figure 2: Learning the conditional distribution for one of the blobs dataset $r_3 = 0.75$. From the top left to the bottom right: the Popularity based sampling, the Selfplay model (top left) the Softmax and the CIS model. Again, on this image, we see the impact of the CIS model on the learning of the right conditional distribution as the distinction between true data (inside the blobs) and negative data is clearer.

| Method | AIS | FastText | Full Softmax |
|---|---|---|---|
| animals | bacteria, humans beings, structures organisms, plants | animal, animalia mammals, humans vertebrates, carnivores | methods, species techniques, practices things, plants |
| brother | son, grandson daughter, father mother, uncle | brothers, broth brotherhood, father father, mother | son, b countess, maria prince, nephew |
| charles | robert, henry edward, elizabeth peter, james | charley, charleston charlie, charlotte frederick, louise | henry, erasmus peter, thomas james, john |
| constantinople | monarch, prussia ruins, athens prague, antioch | constantine, constantinus constantin, adrianople nicaea, chalcedon | carthage, emperors reformation, dynasty ks, bohemia |
| freedom | hate, sociology acceptance, resistance conscious, subject | freedoms, freed freedmen, edom liberty, free | free, learning fairness, anymore huygens, technology |

Table 8: Table showing results for the Similarity task results : in some cases we can see that the CIS model does reflect more the meaning of the word.

