# OpenReview forum: "Partially Mutual Exclusive Softmax for Positive and Unlabeled data"
_ICLR.cc/2019/Conference_

### Official Review · AnonReviewer1 · 2018-11-01
**Interesting idea, writing needs a lot of improvement**

**Rating:** 5
**Confidence:** 4

**Review:**

This paper presents Partially Mutual Exclusive Softmax (PMES), a relaxation of the full softmax that is commonly used for multi-class data. PMES is designed for positive-unlabeled learning, e.g., language modeling, recommender systems (implicit feedback), where we only get to observe positive examples. The basic idea behind PMES is that rather than considering all the non-positive examples as negative in a regular full softmax, it instead only considers a "relevant" subset of negatives. Since we actually don't know which of the negatives are more relevant, the authors propose to incorporate a discriminator which attempts to rate each negative by how hard it is to distinguish it from positives, and weight them by the predicted score from the discriminator when computing the normalizing constant for the multinomial probability. The motivation is that the negatives with higher weights are the ones that are closer to the decision boundary, hence will provide more informative gradient comparing to the negatives that are further away from the decision boundary. On both real-world and synthetic data, the authors demonstrate the PMES improves over some other negative sampling strategies used in the literature.

Overall the idea of PMES is interesting and the solution makes intuitive sense. However, the writing of the paper at the current stage is rather subpar, to the extend that makes me decide to vote for rejection. In details:

1. The motivation of PMES from the perspective of mutual exclusivity is quite confusing. First of all, it is not clear to me what exactly the authors mean by claiming categorical distribution assumes mutual exclusivity -- does it mean given a context word, only one word can be generated from it? Some further explanation will definitely help. Further more, no matter what mutual exclusive means in this context, I can hardly see that PSME being fundamentally different given it's still a categorical distribution (albeit over a subset).

The way I see PMES from a positive-unlabeled perspective seems much more straight-forward -- in PU learning, how to interpret negatives is the most crucial part. Naively doing full softmax or uniform negative sampling carry the assumption that all the negatives are equal, which is clearly not the right assumption for language modeling and recommender systems. Hence we want to weight negatives differently (see Liang et al., Modeling user exposure in recommendation, 2016 for a similar treatment for RecSys setting). From an optimization perspective, it is observed that for negative sampling, the gradient can easily saturate if the negative examples are not "hard" enough. Hence it is important to sample negatives more selectively -- which is equivalent to weighting them differently based on their relevance. A similar approach has also been explored in RecSys setting (Rendle, Improving pairwise learning for item recommendation from implicit feedback, 2014). Both of these perspectives seem to offer more clear motivation than the mutual exclusivity argument currently presented in the paper.

That being said, I like the idea of incorporating a discriminator, which is something not explored in the previous work.

2. The rigor in the writing can be improved. In details:

* Section 3.3, "Multivariate Bernoulli" -> what is presented here is clearly not multivariate Bernoulli

* Section 3.3, the conditional independence argument in "Intuition" section seems no difference from what word2vec (or similar models) assumes. The entire "Intuition" section is quite hand-wavy.

* Section 3.3, Equation 4, 5, it seems that i and j are referred both as binary Bernoulli random variables and categorical random variables. The notation here about i and j can be made more clear. Overall, there are ambiguously defined notations throughout the paper.

* Section 4, the details about the baselines are quite lacking. It is worth including a short description for each one of them. For example, is PopS based on popularity or some attenuated version of it? As demonstrated from word2vec, a attenuated version of the unigram (raised to certain power < 1) works better than both uniform random, as well as plain unigram. Hence, it is important to make the description clear. In addition, the details about matrix factorization experiments are also rather lacking.

3. On a related note, the connection to GAN seems forced. As mentioned in the paper, the discriminator here is more on the "cooperative" rather than the "adversarial" side.

Minor:

1. There are some minor grammatical errors throughout.

2. Below equation 3, "\sigma is the sigmoid function" seems out of the context.

3. Matt Mohaney -> Matt Mahoney

---

> ### Author Response · Authors · 2018-11-09
> **Clarification of the drawbacks of the softmax formulation and the advantages of PMES**
>
> Dear reviewer,
>
> Thank you for your detailed feedback, please find our answers below:
>
> To begin, as a general answer to your feedback, we would like to say that indeed, one can see our PMES as new negative sampling scheme. It enables us to sample true negatives, close to the decision boundary, that will be informative in terms of gradients. Therefore, instead of choosing random and easy negatives as with Uniform Sampling or just all the potential targets as with Softmax, we now have a better strategy for sampling negatives.
>
> However, the difference with previous negative sampling approaches is that we are not trying to approximate full softmax which is the case of all prior work since the time that Sampled Softmax was introduced by Bengio et al in “Quick Training of Probabilistic Neural Nets by Importance Sampling,” where the estimator has been seen as a biased approximation of the full softmax.
>
> In our case, we argue that sampled softmax is ideal because it relaxes the mutual exclusivity constraint and with a good sampling can outperform the full softmax.
> In the case of multi-class and single-label tasks it is natural to use the softmax formulation. However, when it comes to language modelling and sequence prediction, most of the tasks fall in the multi-labeled settings. For a given context, one does not observe one target item j, but rather a subset of target items{j1,...,jk}. For example, in a word2vec setting, the subset of targets is defined by the sliding window parameter.
> Inspired from textual analysis, Blei et al. (2003) (Latent Dirichlet Allocation) suggested that words in sequences can be regarded as a mixture of distributions related to each of the different categories ofthe vocabulary, such as "sports" and "music". Building upon this example, we effectively search over an enlarged class of models to better represent the multiplicity of the data. We now train a product of independent distributions to learn this set generative process.
> To clarify our point, we added a new paragraph in our paper.
>
> Now, going through the different points raised:
> Q1 : The multivariate-Bernoulli formulation has been removed. We prefer now to say that we model the data with a n-dimensional random vector of Bernoulli random variables.
>
> Q2 : Thanks for your comment on this section, the "intuition" paragraph has been edited for clarity.
>
> Q3 : Notations have been clarified in the paper. In the PME Softmax model, one tries to fit parameters of Bernoulli distributions. In section 3.3, i and j refer indeed to binary Bernoulli random variables with parameter P(j|i).
>
> Q4: A short description of each baseline has now been added to the paper. For the popularity sampling, we used a log uniform distribution as used in the TensorFlow implementation.
>
> To be noted the relation to GAN as reduced, at least in the Related Work section. As mentioned earlier, both the generator and the discriminator work in a cooperative setting rather than an adversarial one.
>
> Hope these details improved the understanding of our work,
> Many regards

---

### Official Review · AnonReviewer3 · 2018-11-06
**missing critical details in formulation and evaluation**

**Rating:** 4
**Confidence:** 4

**Review:**

This paper proposed PMES to relax the exclusive outcome assumption in softmax loss. The proposed methods is motivated from PU settings. The paper demonstrate its empirical metrit in improving word2vec type of embedding models.

- on experiment:
-- word2vec the window size = 1 but typically a longer window is used for NS. this might not reflect the correct baseline performance. is the window defined after removing rare words? what's the number of NS used? how stop words are taken care of?
-- would be good to elaborate how CIS in word similarity task were better than full softmax. Not sure what;s the difference between the standard Negative sample objective. Can you provide some quantitative measure?
-- what is the evaluation dataset for the analogy task?

-- MF task: the results/metrics suggests this is a implicit [not explicit (rating based)] task but not clearly defined. Better to provide - embedding dimensions, datasets positive/negative definition and overall statistics (# users, movies, sparsity, etc), how the precision@K are calculated, how to get a positive label from rating based dataset (movielens and netflix), how this compares to the plain PU/implicit-matrix factorization baseline. How train/test are created in this task?


- on problem formulation:
in general, it is difficult to parse the technical contribution clearly from the current paper.
-- in 3.3., the prob. distribution is not the standard def of multi-variate bernoulli distribution.
-- (6) first defined the support set but not clear the exact definition. what is the underlying distribution and what is the support for a sington means?
-- it is better to contrast against the ns approximation in word2vec paper and clarify the difference in term of the mathematical terms.

---

> ### Author Response · Authors · 2018-11-09
> **Explanations on Experiments and Problem formulation**
>
> Dear reviewer,
>
> First of all, thank you for the precise feedback. We 'll try and answer all the different points made above.
>
> On experiments:
> Q1 : Indeed, the window size will impact the quality of the models, but we want to be clear that the same window size has been used for all negative sampling schemes. Indeed, we will undertake further experiments to confirm that the same results are observed on different window sizes. Also, the window is defined after removing the rare words. In the dataset from Matt Mahoney's web page, the stop words are already taken care of. We added more details on this to the updated version of the paper.
>
> Q2 : We are currently working on quantitative experiments for the similarity task. It will indeed help us better differentiate the CIS performance from full softmax. Results will be then added to the paper.
>
> Q3 : The test set used is the Google analogy test set developed by Mikolov et al. It can be found here: http://download.tensorflow.org/data/questions-words.txt
>
> Q4 : For all our experiments, we hold out 20% of the dataset for test time. Indeed, we ran experiments on an implicit task with positive only data. The MPR refers to the standard Mean Percentile Rank and, Prec@k refers to the fraction of instances in the test set for which the target falls within the top-k predictions (we added definitions for both metrics).
> Regarding the movie datasets creation, we only kept movies ranked over respectively 4 and 4.5 stars. From these, we create positive only co-occurence datasets from all the possible pair combinations of items.
> In terms of performance, no experiments have been done to compare sampling based methods and plain implicit-matrix factorization baselines on this dataset. However, many papers in the recent years have underlined the fact that sampling schemes methods can be interpreted as implictly factorizing a word context matrix (Neural Word Embedding as Implicit Matrix Factorization, Levy et al). All these details have been made clearer in the current version of the paper.
>
> On problem formulation:
> Q1 : The multivariate-Bernouilli formulation has been removed. We prefer to say that we model the data with a n-dimensional random vector of Bernoulli random variables.
>
> Q2 : The support set Si is defined as all the potential targets j that can possibly co-occur with i. Si is therefore defined as a subset of J. The definition has been clarified in the paper.
>
> Q3 : We did not compare with the NS formulation of Mikolov paper, Distributed Representations of Words and Phrases and their Compositionality, as we are not using the same loss. However, NS as defined by Mikolov does not try to fit a generative model and therefore does not fall within the scope of our PME Softmax. Further experiments could try our CIS sampling scheme to Mikolov's NS loss to see if it improves the performance.
>
> Hope these details improved the understanding of our work,
> Best regards.

---

### Official Review · AnonReviewer4 · 2018-11-13
**Interesting idea, need more clarification and detail, not sure if language modeling is good application**

**Rating:** 5
**Confidence:** 4

**Review:**

The mutually exclusive assumption of traditional softmax can be biased in case negative samples are not explicitly defined. This paper presents Cooperative Importance Sampling towards resolving this problem. The authors experimentally verify the effectiveness of the proposed approach using different tasks including applying matrix factorization in recommender system, language modeling tasks and a task on synthetic data.

I like this interesting idea, and I agree with the authors that softmax does exist certain problem especially when negative samples are not well defined. I appreciate the motivation of this work from the PU learning setting. It would be interested to show more results in PU learning setting using some synthetic data. I am interested to see the benefit of this extension of softmax with respect to different amount of labeled positive samples.

However, I am not completely convinced that the proposed method would be a necessary choice for language modeling tasks.
--To me, the proposed method has close connection to 2-gram language model.
--But for language tasks, and other sequential input, we typically make prediction based on representation of very large context. Let’s say, we would like to make prediction for time step t given the context of word_{1:t} based on some recurrent model, do you think the proposed softmax can generally bring sizable improvement with respect to traditional choices. And how?

By the way, I think the proposed method would also be applicable in the soft-label setting.

For the experiments part, maybe put more details and discussions to the supplementary material.
A few concrete questions.
-- In some tables and settings, you only look at prec@1, why? I expect the proposed approach would work better in prec@K.
-- Can you provide more concrete analysis fortable 6? Why proposed methods does not work well for syntactic.
-- Describe a little bit details about MF techniques and hyper-parameters you used.

---

### Meta-Review · Area_Chair1 · 2018-12-18
**not above threshold**

**Confidence:** 4
**Recommendation:** Reject

**Metareview:**

All reviewers agree that the paper is not quite ready for publication.